# Variable Selection in Untargeted Metabolomics and the Danger of Sparsity

**DOI:** 10.3390/metabo10110470

**Published:** 2020-11-17

**Authors:** Gerjen H. Tinnevelt, Udo F.H. Engelke, Ron A. Wevers, Stefanie Veenhuis, Michel A. Willemsen, Karlien L.M. Coene, Purva Kulkarni, Jeroen J. Jansen

**Affiliations:** 1Institute for Molecules and Materials, Radboud University, 6525 AJ Nijmegen, The Netherlands; jj.jansen@science.ru.nl; 2Translational Metabolic Laboratory (TML), Department of Laboratory Medicine, Radboud University Medical Center, 6525 GA Nijmegen, The Netherlands; Udo.Engelke@radboudumc.nl (U.F.H.E.); ron.wevers@radboudumc.nl (R.A.W.); karlien.coene@radboudumc.nl (K.L.M.C.); purva.Kulkarni@radboudumc.nl (P.K.); 3Department of Pediatric Neurology, Radboud University Medical Centre, 6525 GA Nijmegen, The Netherlands; stefanie.veenhuis@radboudumc.nl (S.V.); michel.willemsen@radboudumc.nl (M.A.W.)

**Keywords:** univariate/multivariate statistics, chemometrics, untargeted metabolomics, variable selection, ataxia–telangiectasia, vitamin B3/nicotinamide riboside treatment

## Abstract

The goal of metabolomics is to measure as many metabolites as possible in order to capture biomarkers that may indicate disease mechanisms. Variable selection in chemometric methods can be divided into the following two groups: (1) sparse methods that find the minimal set of variables to discriminate between groups and (2) methods that find all variables important for discrimination. Such important variables can be summarized into metabolic pathways using pathway analysis tools like Mummichog. As a test case, we studied the metabolic effects of treatment with nicotinamide riboside, a form of vitamin B3, in a cohort of patients with ataxia–telangiectasia. Vitamin B3 is an important co-factor for many enzymatic reactions in the human body. Thus, the variable selection method was expected to find vitamin B3 metabolites and also other secondary metabolic changes during treatment. However, sparse methods did not select any vitamin B3 metabolites despite the fact that these metabolites showed a large difference when comparing intensity before and during treatment. Univariate analysis or significance multivariate correlation (sMC) in combination with pathway analysis using Mummichog were able to select vitamin B3 metabolites. Moreover, sMC analysis found additional metabolites. Therefore, in our comparative study, sMC displayed the best performance for selection of relevant variables.

## 1. Introduction

Metabolites are small molecules that are the substrates, intermediates, or end products of metabolism. Alterations in metabolite concentration may be related to a disease or may be caused by xenobiotics [1]. Untargeted metabolomics aims to measure as many metabolites as possible and is often used as a hypothesis-generating or an exploratory tool. One popular analytical technique for untargeted metabolomics is LC/MS, which can generate thousands of features. However, the majority of these features remains unidentified, i.e., their masses do not match any known metabolite present in metabolite databases [2]. Additionally, alterations of a single enzyme may lead to a cascade of secondary metabolic effects, which are measured in untargeted metabolomics but their impact on disease presentation and development is unclear [3].

One goal of untargeted metabolomics is finding metabolite biomarkers predictive of incidence or outcome of disease [4]. Popular chemometric methods for finding biomarkers include univariate analysis such as *t*-tests and also multivariate analysis such as principal component analysis (PCA), partial least squares (PLS), and orthogonal projections to latent structures discriminant analysis (OPLS-DA) [5]. After calculating these multivariate methods, variables may be selected using variable influence on projection (VIP) [6], selectivity ratio (SR) [7], or significance multivariate correlation (sMC) [8]. These methods render many variables of importance. Additionally, sparse methods exist that select variables when creating the model such as lasso [9], sparse PLS (sPLS) [10] and CARS-PLS [11]. These methods find the minimal set of variables needed for an optimal prediction of group membership. All of these methods restrict the number of metabolites included in the differentiating biomarker, where sparse methods have an inherent restriction for a minimal set of contributing metabolites. This complies with Occam’s razor that dictates that models should be simplified as much as necessary. However, this does introduce an apparent contradiction, in which the chemical analysis of metabolomics aims for full metabolite coverage and the data analysis aims for optimal simplification. Within metabolic pathways, considerable correlations may exist between different metabolites, imposed by the number of chemical reactions they are involved in. These three aspects together create a discrepancy in which the data analysis may ignore specific metabolites that are being measured in the chemical analysis, because their levels are so highly correlated with the levels of other metabolites, and that are then included in the model. Ignoring such redundant information will not hamper the predictive power of the models, but will seriously limit the mechanistic information that can be obtained from the valuable metabolomics data. 

After finding a discriminative set of variables, the associated features must be annotated to actual metabolites to put the results in biological context regarding the disease studied. Mummichog [12] is an algorithm that annotates features to metabolic pathways based on their *m/z* ratio and retention time. If a metabolic pathway contains a high number of significant features, this pathway is most likely affected. Mummichog thus requires to find as many true significant features as possible. Sparse discrimination methods are less suited for Mummichog, because these sparse methods select few features and ignore redundant or correlated features and thus will probably only select a single feature from a pathway to discriminate between groups. Univariate analysis was used before by the authors of Mummichog and, although useful, univariate analysis does not consider the covariance structure present in pathways. Potential metabolites may be missed that do not have a significant difference by themselves, but may in combination with other metabolites (e.g., metabolites in the same pathway) be different between groups. The variable selection method sMC is able to find correlating variables with the response while ignoring the effect of irrelevant correlating structures, a known problem of VIP [8]. For this reason, we hypothesized that the combination of Mummichog with sMC would enhance the possibility of discovering altered metabolic pathways. 

In this paper, we compared the performance of multiple variable selection methods in combination with Mummichog for the identification of vitamin B3 metabolites in a cohort of patients with ataxia–telangiectasia (A–T) orally supplemented with vitamin B3 in the form of nicotinamide riboside (for the rationale and study protocol, see: www.clinicaltrials.gov, identifier: NCT03962114). Vitamin B3 is an important co-factor for many cellular processes, including fatty acid metabolism and energy metabolism [13]. Therefore, supplementing vitamin B3 is expected to alter the concentration of many metabolites alongside increasing the levels of vitamin B3 metabolites. The goal of this untargeted metabolomics study was to see whether the vitamin B3 pathway-related metabolites were indeed increased and which other metabolites and/or pathways were influenced upon treatment. This study is thus an excellent showcase for the discrepancy between data analysis (optimal simplification) and untargeted metabolomics (full metabolite coverage). 

## 2. Results

The results are subdivided into four sections. Section 2.1. shows that during the treatment the vitamin B3 metabolites were indeed increased with a high fold change, a low *p*-value, and a high sMC F-value. Section 2.2. indicates that all methods could easily discriminate between samples taken before and during treatment. Section 2.3. shows that Mummichog could indeed find the vitamin B3 pathway and additional pathways. Finally, Section 2.4. describes correlation analysis that found relationships between metabolites that were missed by pathway analysis. Metabolites assigned by Mummichog are the best possible guesses based on their *m/z* values, and their name was only given in the main text when confirmed by measurement of a model compound in the Ultra-High Performance Liquid Chromatography-Quadrupole Time-Of-Flight Mass Spectrometry (UHPLC-QTOF-MS) setup [14]. The only exception isN1-methyl-4-pyridone-5-carboxamide and N1-methyl-2-pyridone-5-carboxamide.

### 2.1. Vitamin B3 Pathway-Related Metabolic Features

Nicotinic acid (**1**) degrades via nicotinamide (**2**) into N1-methyl-4-pyridone-5-carboxamide (**3**) or N1-methyl-2-pyridone-5-carboxamide (**4**), see Table 1. Mummichog assigned two features at a weight tolerance of 1 ppm to nicotinic acid or picolinic acid as isobaric metabolites. Nicotinic acid and picolinic acid were both detected according to the Human Metabolome Database (HMDB) [15]. Three metabolic features were assigned to nicotinamide (**2**) at a weight tolerance of 1 ppm. Metabolites **3** and **4** are isomers, and the five features at 1 ppm were assigned to both metabolites. At a weight tolerance of 5 ppm, two more metabolic features were assigned to **1**, five more to **2**, and 23 metabolic features were assigned to metabolites **3** and **4**. Most of these features had different retention times than the features found at 1 ppm except for the M+K[1+] adduct at a retention time of 2.86 minutes and M+CH3COO[−] adduct at 2.97 minutes for metabolite **3** or **4**. Mummichog probably assigns many false positives at 5 ppm and misses some (false negatives) at 1 ppm. Most features and metabolites in Table 1 had a significantly higher intensity after treatment. Although no definitive level 1 identification [16] of these four metabolites could be given, it is highly likely that they indeed represent vitamin B3-related metabolites, increased upon treatment with nicotinamide riboside, also because the *p*-value is low and the sMC F-value is extremely high. A high sMC F-value means the feature explained much variance of the response to treatment compared to other structures in the data. 

### 2.2. Treatment of A–T Patients with Nicotinamide Riboside

Table 2 shows that all tested methods could clearly distinguish the patient samples before and during treatment with nicotinamide riboside. The univariate Wilcoxon sign test and multivariate methods OPLS-DA with VIP and Weight Randomization Test for partial least squares (WRT-PLS) with sMC render hundreds of deviating features, whereas the sparse methods CARS-PLS-DA and sPLS-DA only result in a handful of altered features. A feature with *m/z* 87.0089 (C_3_H_4_O_3_ M − H[−]) was increased more than 47-fold upon treatment. Mummichog assigned this feature to eight metabolites and, of the total fifteen features that CARS-PLS-DA found, only two others were assigned to known metabolites. The same features were also found with sPLS-DA, but additionally, eleven more features could be annotated including the M + Cl[−] of metabolite **3** or **4**. All of the 48 selected features had a high positive (>0.63) or high negative (<-.55) correlation with the features of Table 1. Only eight features had an sMC F-value higher than 187.52 and only two metabolic features had an sMC F-value higher than 539.53. Thus, the sparse methods CARS-PLS-DA and sPLS-DA did not only select the features with the highest discriminatory power, see Figure 1. 

### 2.3. Pathway Analysis with Mummichog Based on Features Selected with Wilcoxon or sMC

The feature table used for analysis contained a total of 22,684 features. Table 3 shows the number of features assigned to metabolites by Mummichog. Next, Mummichog was used to find significant enriched pathways based on the Wilcoxon sign test and sMC. Pathways may have too many false-positive features assigned to them and permutation testing is used to detect those. The *p*-values were randomly permutated 100 times, each time the significance level was calculated for the pathway, and the part of permutations in which the pathway had a lower significance level than the actual data was calculated. The permutation test results of the vitamin B3 pathway can be seen in Table 3, which shows that only at a weight tolerance level of 1 ppm the pathway is not a false positive, regardless of the approach. 

The additional pathways identified by Mummichog are listed in Table 4. The analysis based on the *p*-values derived from the Wilcoxon sign test only found the vitamin B3 pathway to be significantly altered, whereas sMC identified additional significant pathways. The metabolic features associated with metabolites of the arachidonic acid pathway were decreased during nicotinamide riboside treatment. Another pathway found with sMC at α = 0.01 involved methionine and cysteine metabolism. Adenosine and another metabolite were significantly increased, one metabolite was significantly decreased, and five metabolites showed a significant multivariate correlation with response to the treatment, including cysteine. 

### 2.4. Correlation Analysis with Only the Significant Annotated Metabolites

Finally, pathway analysis does not show any relationship between the vitamin B3 metabolites and other metabolites. For this reason, the correlation matrix was calculated between all significant features annotated with primary ions. An absolute correlation of 0.8 was used as cut-off to find structures in the data. Most features were uncorrelated except for one highly connected network containing 21 features, including the vitamin B3 metabolites. The following assigned metabolites in the network were significantly increased upon treatment with nicotinamide riboside: nicotinamide, nicotinic acid, N1-methyl-4-pyridone-5-carboxamide, N1-methyl-2-pyridone-5-carboxamide, hypoxanthine, guanosine, inosine, adenosine, and two more. Four assigned metabolites were significantly decreased. The purine metabolism pathway was not significant in the pathway analysis because many non-significant features were also assigned to metabolites from this pathway. 

## 3. Discussion 

Untargeted metabolomics with LC/MS identifies thousands of features in patient body fluids. Adequate variable selection is therefore essential to find clinically relevant features. In our study, a cohort of patients with A–T was sampled before and during treatment with nicotinamide riboside. The goal of this study was to compare multiple variable selection methods on their ability to find the nicotinamide riboside-related metabolites and how treatment with vitamin B3, as an important co-factor in human metabolism, would influence other metabolic pathways. All the compared variable selection methods could easily distinguish between before and during treatment patient samples, indicating that the individual variability is much smaller than the difference between before and during treatment. The sparse methods CARS-PLS-DA and sPLS-DA were used to find the minimal set needed for optimal discrimination between before and during treatment. Wilcoxon signed test and partial least squares in combination with VIP and sMC were used to find all discriminatory features. Although no metabolic feature was assigned to nicotinamide riboside, features were assigned to vitamin B3 metabolites. The reason that we do not observe nicotinamide riboside itself as increased may be found in matrix-related ion suppression, as we were able to detect the pure compound when dissolved in water. Another reason could be that nicotinamide riboside itself is metabolized rapidly to the associated metabolites that we do assign.

The vitamin B3-related features were significantly increased and had a low Wilcoxon signed test *p*-value, high fold change, high VIP, and high sMC. However, sparse methods only need a subset of those features and thus may only find features that correlate to the feature of interest. For example, CARS-PLS-DA did not find any metabolic feature related to the vitamin B3 pathway and sPLS-DA only found one metabolic feature. The found features did have a high correlation with the vitamin B3 metabolic features. Moreover, not all found features had the highest multivariate correlation significance. These sparse methods probably use some high discriminatory features to discriminate and some less discriminatory features to stabilize the multivariate space. Variable selection in combination with random forest or support vector machines was briefly explored (results not shown), but we did not find any deterministic results and neither did variable selection based on genetic algorithms, because there are multiple subsets possible of the hundreds of discriminatory features. 

Another challenge in untargeted metabolomics is the annotation of features to metabolites. Mummichog can automatically assign features to metabolites that are in known metabolic pathways. At a molecular weight tolerance of 1 ppm, Mummichog only assigned 4.55% of all features. Using a higher molecular weight tolerance leads to more assignments, but also increases the rate of false-positive assignments. Mummichog automatically assigns all features to metabolites based on their *m*/*z* value. Mummichog thus considers the ppm value and the number of adducts that were also assigned to the same metabolite [12]. For the metabolites specified in this paper, their identity has been confirmed by measurement of a model compound in the UHPLC-QTOF-MS setup [14]. For all other metabolites, such as the arachidonic acid pathway metabolites, further confirmation of identity is still needed. Our approach should therefore be considered as a hypothesis-generating tool, rendering interesting leads for biochemical follow-up studies. 

Pathway analysis of the effects of nicotinamide riboside treatment using sMC as input for the Mummichog algorithm pointed to vitamin B3, arachidonic acid, and methionine and cysteine metabolism to be affected. Metabolites of arachidonic acid were decreased and vitamin B3 is known to inhibit the breakdown of lipids [17]. In another study, methionine and cysteine were found to increase in Sprague Dawley rats upon vitamin B3 intake during three months [18]. However, methionine was not found to be significantly different and cysteine was found only multivariately upon treatment with nicotinamide riboside in patients with A–T. The pathway reference database MetaFishNet is a compilation of multiple databases including the Kyoto Encyclopedia of Genes and Genomes (KEGG) database [19,20,21]. Most of the significant features were not included in the methionine and cysteine pathway of the KEGG database. Thus, if just the KEGG database is used for the reference of pathways, methionine and cysteine metabolism would not be significant. In other words, pathway analysis does not consider the importance of certain metabolites inside a pathway. The core metabolites are not necessarily significantly different as seen with the methionine and cysteine pathway. Finally, changing the significance level of sMC to α = 3 × 10^−7^ found the same number of metabolites as the Wilcoxon signed rank test for the arachidonic acid pathway, but because the total number of significant metabolites was lower, arachidonic acid was a significant pathway when using sMC compared to the Wilcoxon signed rank test. Thus, finding more important features is of less value if these features are not biochemically related and sMC may help to find many correlated features. 

## 4. Materials and Methods

### 4.1. Measurements and Conversion to the Feature Intensity Table

Plasma samples from 14 patients with A–T before treatment and during treatment with vitamin B3 (nicotinamide riboside) were measured using UHPLC-QTOF-MS in both positive and negative ionization mode according to a previously described procedure [14]. Considering the processing of raw QTOF-MS data, the vendor data format (.d) was converted into open format .mzML data using MSConvert [22]. Data files were preprocessed, which included alignment, peak picking, and grouping using XCMS [23]. All the data conversion and preprocessing steps were performed as a part of an in-house developed bioinformatics pipeline. All patient samples were measured in duplicate, and an internal quality control (QC) sample was also included every 9th or 10th measurement. The use of patient samples for this study was approved by the Regional Committee on Research involving Human Subjects Arnhem-Nijmegen (NL68197.091.18). Informed consent was obtained from all patients and/or their legal caregivers according to the tenets of the Declaration of Helsinki.

### 4.2. Additional Preprocessing of the Large Feature Intensity Table

The preprocessed output encompasses a large feature intensity table where each feature contains an *m/z* value, a retention time, and an intensity (relative abundance) for every measured sample. Features with more than 20% zeroes (missing values) in the samples or with any zero values in the QC samples were removed. Features with an *m/z* value lower than 70 or higher than 700 were removed. Features with a retention time lower than 0.4 minutes or higher than 16 minutes were removed. Intra-batch correction was performed using the internal QC samples and support vector regression (QC-SVR) [24]. Other preprocessing steps included Probabilistic Quotient Normalization (PQN) using the median of internal QC samples as a reference [25], KNN imputation (k = 10, Euclidian distance) of the zero values (missing values), glog transformation optimized based on the internal QC samples [26], and mean centering (no scaling) [27]. The rationale behind PQN is that the majority of (housekeeping) metabolites do not change. The quotients of most samples were close to 1, meaning that the difference between samples was very small and minimized during the sample preparation. Pareto scaling was explored, but standard PLS was unable to distinguish between before and during treatment patient samples; therefore, no scaling was used. Matlab functions were written and used using Mathworks Matlab R2020a. 

### 4.3. Data Analysis, Variable Selection, and Validation

From each duplicate measurement, the mean was calculated prior to univariate analysis with a Wilcoxon signed rank test. Multiple testing correction was performed using the false discovery rate (FDR) approach of Benjamini–Hochberg [28]. For multivariate analysis, the duplicate measurements were used directly, because multivariate methods may distinguish between variance that explains the response and variance that does not explain the response, e.g., instrumental variability. Multivariate analysis included the following: OPLSDA [29,30] in combination with VIP [6], weight randomization test for partial least squares (WRT-PLS) [31] in combination with sMC [8], CARS-PLS-DA[11] with two latent variables, and sparse PLS-DA with one latent variable [10]. Double cross-validation was used to validate the results [32]. Five-fold cross-validation with ten iterations was used for the outer loop to determine the accuracy. Four-fold cross-validation was used for the inner loop to determine the number of latent variables in OPLS-DA and the variables in CARS-PLS-DA and sparse PLS-DA. Inner cross-validation was not required in WRT-PLS. Variables were selected based on the training set by VIP, sMC, CARS-PLS-DA, and sparse PLS-DA, and then a new PLS-DA model was made using only those variables and the training set, see Figure 2. Accuracy was calculated based on the prediction of the test set. Features that were selected during all cross-validation iterations were listed per method. Matlab functions were written and used using Mathworks Matlab R2020a.

### 4.4. Annotation and Pathway Analysis with Mummichog

Mummichog v2.0 was used along with the MS Peaks-to-Pathway module available on Metaboanalyst v4.0 (https://www.metaboanalyst.ca) [12,33]. The input file requires *m/z* values, retention time values, *p*-values, and information on whether the feature is acquired from negative or positive ionization mode. The sMC F values were converted to *p*-values and were used, as well as the *p*-values of the univariate test. The molecular weight tolerance was used at 1, 3, and 5 ppm together with the option to enforce primary ions. Only features with at least one assigned primary ion (M + H[1+], M + Na[1+], M − H_2_O + H[1+], M − H[1−], M − 2H[2−], M − H_2_O − H[1−]) were valid. The *p*-value cut-off was set to α = 0.01 or 3 × 10^−7^ for sMC and to 0.0012 (FDR-corrected α = 0.05) for univariate testing. The human reference pathways of the MetaFishNet database were used for pathway analysis in Mummichog [34].

## 5. Conclusions

Through untargeted metabolomics, hundreds of altered metabolic features were identified in ataxia–telangiectasia (A–T) patients upon treatment with nicotinamide riboside, which included vitamin B3-associated metabolites. Chemometric methods can easily distinguish between samples obtained before and during treatment; however, assigning all the metabolic features to metabolites and connecting them to disease mechanisms is still a challenging task. Moreover, the definition of a limited set of key metabolites to monitor therapy response is preferred in light of future application in clinical diagnostics. 

For this reason, sparse methods were tested that focus on finding a minimum set of metabolic features to discriminate between patient samples before and during treatment. However, these sparse methods did not find vitamin B3 metabolites but only found features that correlate with the vitamin B3 metabolites. Another approach is to summarize the detected metabolites into their related pathways using pathway analysis. Mummichog in combination with *p*-values obtained from univariate testing only found the vitamin B3 pathway. Using Mummichog with significant multivariate correlation based on partial least squares discriminant analysis also found the vitamin B3 pathway and additionally found the arachidonic acid pathway. The metabolites related to the arachidonic acid pathway still need further confirmation of their identity, to draw definite conclusions. Additional correlation analysis of the significant annotated features found that the purine nucleosides, i.e., adenosine, inosine, and guanosine, correlate to the vitamin B3 metabolites. In conclusion, our data show that treatment with nicotinamide riboside in patients with A–T leads to an increase of vitamin B3 metabolites in plasma as expected, and also increases plasma purine nucleosides and may decrease metabolites in the arachidonic acid pathway. Even though we realize that many metabolic features that may provide additional relevant information still remain unannotated, we show that significant multivariate correlation in combination with pathway analysis and correlation analysis is able to render more leads for putatively affected biochemical processes as compared to sparse methods.

## Figures and Tables

**Figure 1 metabolites-10-00470-f001:**
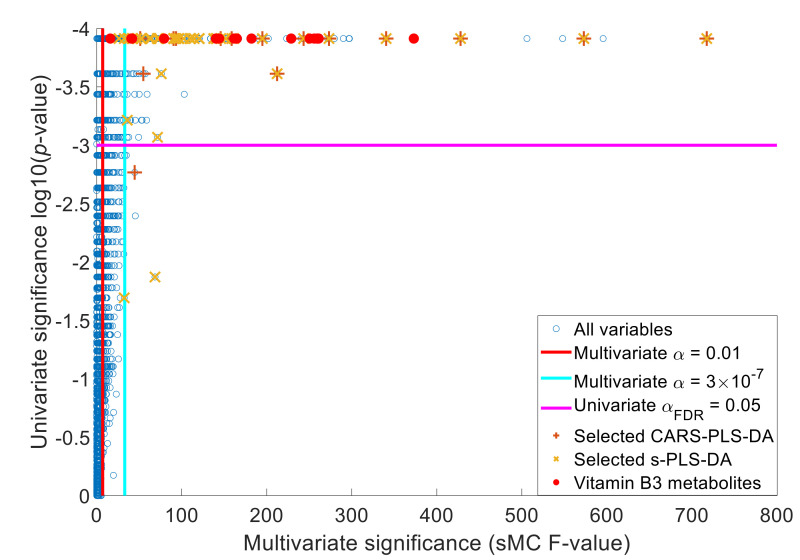
Multivariate significance (sMC F-value) against the univariate significance log10(*p*-value) of the variables (blue unfilled circles). The vitamin B3 metabolites (red filled circles) were both multivariately and univariately significantly different between before and during treatment. Despite selecting many significant variables, the sparse variable selection methods sPLS-DA (yellow crosses (×)) and CARS-PLS-DA (brown plus signs (+)) did not select the vitamin B3 metabolites.

**Figure 2 metabolites-10-00470-f002:**
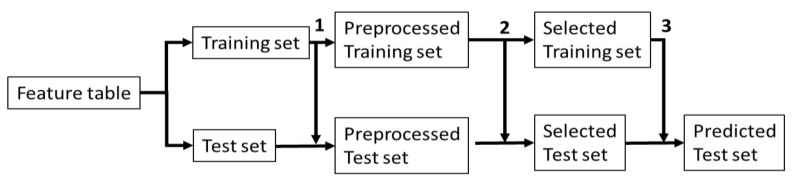
Schematic representation of cross-validation procedure. First, the data were divided into a training set and a test set. (**1**) Preprocessing of data using parameters derived from the training set such as mean and k neighbors. (**2**) Inner cross-validation based on the training set to select variables and number of (orthogonal) latent variables using DQ^2^. (**3**) PLS-DA model based on the variable selected training set and used to predict the test set. Based on the predicted test set, an accuracy was calculated.

**Table 1 metabolites-10-00470-t001:** Assigned metabolites of vitamin B3 metabolism: the degradation of nicotinic acid (**1**) into nicotinamide (**2**) and into N1-methyl-4-pyridone-5-carboxamide (**3**) or N1-methyl-2-pyridone-5-carboxamide (**4**) at a weight tolerance of 1 ppm using Mummichog.

Assigned Metabolite	Adduct	*m*/*z*	Retention Time	Fold Change	Wilcoxon *p*-Value	sMC F-Value
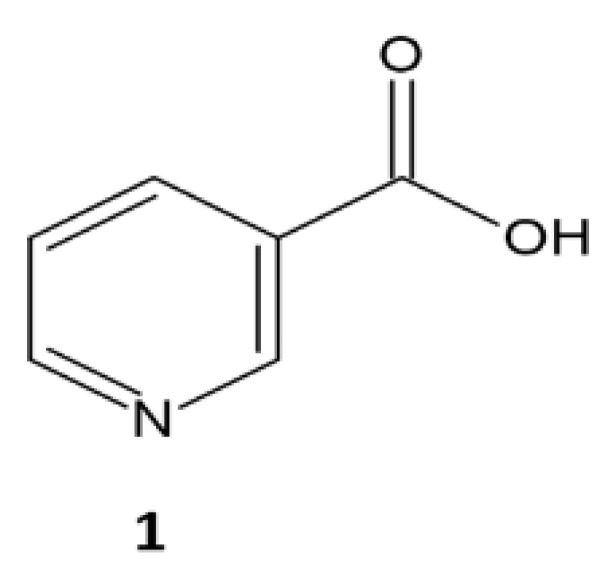	M + H[1+]	124.0329	1.20	7.00	1.2 × 10^−4^	302.90
	M + H[1+]	124.0329	1.01	2.02	1.2 × 10^−4^	118.44
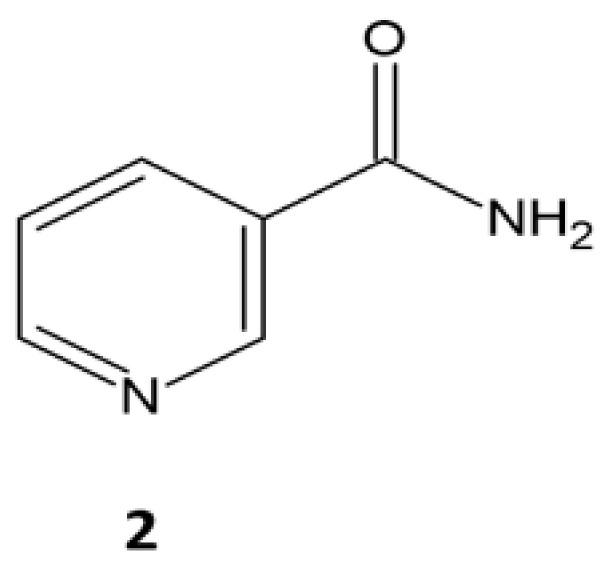	M − H_2_O + H[1+]	105.0445	3.52	1.06	0.9	0.0149
	M + H[1+]	123.0552	1.38	2.76	1.2 × 10^−4^	193.84
	M + H[1+]	123.0553	0.97	1.94	1.2 × 10^−4^	254.34
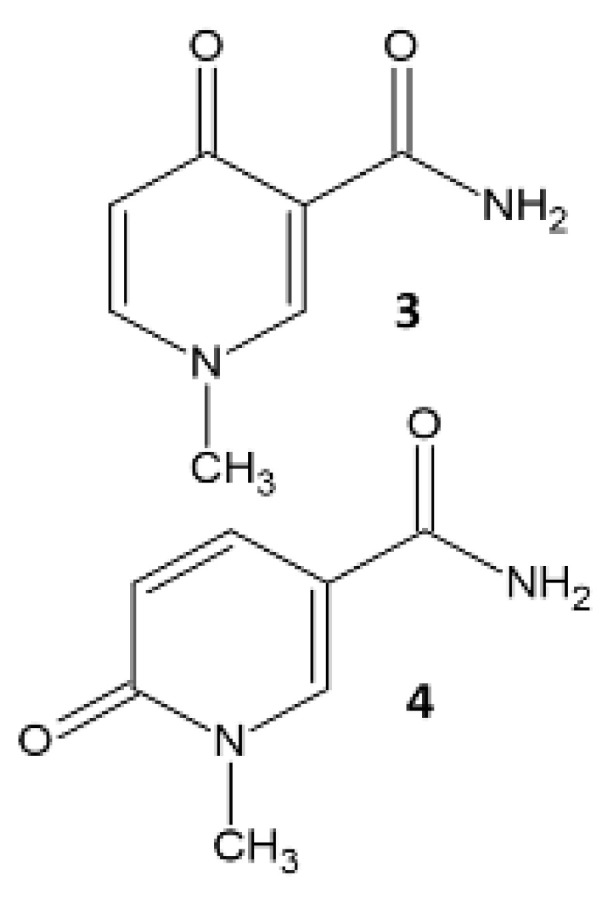	M−NH_3_ + H[1+]	136.0394	2.86	9.50	1.2 × 10^−4^	361.59
	M − H[−]	151.0511	2.97	6.77	1.2 × 10^−4^	359.63
	M + H[1+]	153.0659	2.96	7.75	1.2 × 10^−4^	377.00
	M + Na[1+]	175.0479	2.86	6.98	1.2 × 10^−4^	539.53
	M + Cl[−]	187.0276	2.97	7.20	1.2 × 10^−4^	187.52

**Table 2 metabolites-10-00470-t002:** Overview of results of different methods to distinguish patients with A–T before and during treatment with nicotinamide riboside.

Method	Wilcoxon Sign Test with False Discovery Rate Correction	OPLS-DA VIP	WRT-PLS sMC α = 0.01 F > 7.06	WRT-PLS sMC α = 3 × 10^−7^ F > 32.99	CARS-PLS-DA	sPLS-DA
Accuracy	N/A	100%	100%	100%	100%	100%
Number features	612	842	770	214	15	48
Annotated features	51	41	73	23	3	14

The last row of the table shows the number of metabolites annotated by Mummichog.

**Table 3 metabolites-10-00470-t003:** Mummichog annotation results and vitamin B3 pathway permutation results.

Tolerance	Features	Metabolites	Ratio	Wilcoxon Sign Test	sMC α = 0.01 F > 7.06	sMC α = 3 × 10^−7^ F > 32.99
1 ppm	1033	618	2.41	0	0	0
3 ppm	2465	959	2.68	0.99	1	0.46
5 ppm	3268	1131	2.68	1	1	0.88

The column Features shows the number of features that could be assigned to a metabolite. The column Metabolites shows the number of unique metabolites detected. The column Ratio shows the average number of metabolites assigned to the same metabolic feature. The columns Wilcoxon sign test, sMC α = 0.01, and α = 3 × 10^−7^ show the permutation value for the vitamin B3 pathway.

**Table 4 metabolites-10-00470-t004:** Pathways found by Mummichog at a weight tolerance of 1 ppm.

Pathway	Wilcoxon Sign Test	sMC α = 0.01	sMC α = 3 × 10^−7^
Vitamin B3	5 *	5 *	4 *
Arachidonic acid	3	6 *	3 *
Methionine and cysteine	4	9 *	2

The table lists the number of significant metabolites found within the pathway. The asterisk (*) means that it is significant according to the more conservative Expression Analysis Systematic Explorer (EASE) test.

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
