# Peer review of "Variable Selection in Untargeted Metabolomics and the Danger of Sparsity"

_metabolites, 2020, doi:10.3390/metabo10110470_

Round 1

Reviewer 1 Report

The reviewer really enjoyed the paper and thinks it was a great study. I have no fundamental issues with the science or data analysis. The reviewer appreciated the comparison of ppm thresholds to highlight miss assignment of features, which is helpful to the metabolomics community. My only issues are with presentation and flow of the paper. Overall the paper’s goal seemed to shift throughout reading. On 178 the authors stated “The goal of this study was to find the nicotinamide riboside related metabolites and how treatment 179 with vitamin B3, as an important co-factor in human metabolism, would influence other metabolic 180 pathways.”  The paper doesn’t seem to make this clear since at times there was such a high focus on the comparison of the informatics pipelines. In other words, the Reviewer wasn’t clear what the goal was during review. It seemed to me like the goal was to test informatic algorithms/pipelines on a common dataset. If you’re pitching this as the goal was to find specific metabolites, you should discuss consequences of metabolism, the metabolic pathways, or what the results mean in a biological context. Also there seemed like there was a large build up and focus on sMC, but the authors never really discuss why, which would help readers that are unfamiliar with this method. The authors should take this into consideration and revise as necessary.

  • In the abstract I think “explain” may not be the best word to use. Indicative may be a better choice. What explains disease mechanisms are usually proteo/genomic alterations, metabolites are the downstream consequences of an upstream change.
  • Line 55 consider using a comma between “in within” or rephrasing, I had to re-read the sentence twice.
  • Section 4.2. Why not use Pareto scaling? I saw the data were centered, but why not scaled? Also was why PQN selected?
  • 3. please specify the FDR approach. After reading 4.4 I think I understand, but maybe elaborate why at that section or reference that the information is in 4.4. I initially thought Bonferroni, Benjamini-Hochberg, etc, and wondered why it wasn’t specified
  • 3 why were duplicate measures used in multivariate analysis? This isn’t clear and the Reviewer doesn’t understand.
  • Regarding the arachidonic acid observation on like 154, was nicotinamide riboside administered as a salt with a fatty acid or amino acid?
  • VIP and sMC were already defined. PLS is used here non-defined. Please keep this consistent.
  • The amount of data presented doesn’t seem to mirror the complexity of the experiment and data. Is there anything else that can be presented or maybe a figure that could help get the information across easier?

Reviewer 2 Report

The manuscript by Tinnevelt G.H. et al. compares data analysis methods for their ability to extract meaningful data from untargeted metabolome analysis. The manuscript was interesting to read and seems to be original research with no major flaws. Although all the technical details may be somewhat challenging to follow for an average reader the manuscript comes to a clear conclusion, which is simple to understand and should be of interest for most readers.  There are a few suggestions for improving the article though.

  1. The setup of vitamin treatment should be described in more detail. What was the dose and duration, was it intravenous or oral administration and how was the during-treatment blood sampling timed relatively to vitamin administration. These data give a background for the observed changes and their magnitudes as well as help to interpret the pathway analysis outcomes. In related notes, if nicotinamide riboside was given, how come it was not among the detected metabolites? Since ataxia telangiectasia is not a common disease, maybe it is also possible to briefly explain to the readers why such rare disease and vitamin B3 combination was chosen.  
  2. Lines 194-195 in Discussion confuse me as they imply that random forest and support vector machine algorithms were applied to the results at some point, but the Results did not mention it. Additionally, the two “didn’t” in this paragraph should be replaced by “did not”.
  3. Conclusions are comparable to Discussion in their length and their style is more fitting to a Summary. Perhaps it can be shortened?

Reviewer 3 Report

Tinnevelt et al. found that significant multivariate correlations (sMCs) combined with pathway analysis using Mummichog performed best in the selection of relevant variables to capture biomarkers that explain disease mechanisms. It appears to be a powerful tool in untargeted metabolomics because it can identify peaks from the pathway for mechanistic analysis prior to identification of a substance.

However, no matter how good the analysis method is, unreliable information should be eliminated before the analysis. For example, in compound 4 in Table 1, Cl adduct is listed as a candidate because Cl is a characteristic ion with an isotopic ratio of 35 and 37 of about 3:1, which makes it easy to identify. In addition, since vitamin B3 was administered, it was easy to identify the peaks of vitamin B3 and its metabolites by comparing the samples before and after administration. This is more of a drug metabolism analysis than a metabolomics study. Before applying the present method, it should first be mentioned whether it is not necessary to take into account peaks that can be identified by the classical method, which has been used for decades.

Quantification should also be discussed. Quantification is subject to variability depending on the instruments used and the processing of the sample, as well as variability due to individual differences between subjects. In particular, the latter may be influenced by genetic and environmental factors, including circadian rhythms and potential coexistence of other diseases. It should be mentioned to what extent these factors of variation affect the analysis.

There needs to be a bit more explanation as to why arachidonic acid is coming into play. It seems to be a far from vitamin B3. Also, what about other fatty acids, eicosanoids, phospholipids, etc. would need to be mentioned.

Table 3 displays the m/z errors and the number of peaks. Probably the true value has a variation close to the normal distribution. In other words, even with an accuracy of 5 ppm, the reliability of peaks near the center should be different from the reliability of peaks on the edge of 5 ppm. It would be necessary to explain whether such weighting is done at the peak annotation stage.

Round 2

Reviewer 3 Report

Authors clearly addressed the questions.